# Correlation between Hemophilia Early Arthropathy Detection with Ultrasound (HEAD-US) score and Hemophilia Joint Health Score (HJHS) in patients with hemophilic arthropathy

**Marcel Prasetyo**[1]*, **Ratna Moniqa**[1], **Angela Tulaar**[2], **Joedo Prihartono**[3], **Stefanus Imanuel Setiawan**[1]

**1** Department of Radiology, Faculty of Medicine Universitas Indonesia–Dr. Cipto Mangunkusumo National Central General Hospital, Jakarta, Indonesia, **2** Department of Medical Rehabilitation, Faculty of Medicine Universitas Indonesia–Dr. Cipto Mangunkusumo National Central General Hospital, Jakarta, Indonesia, **3** Department of Community Medicine, Faculty of Medicine Universitas Indonesia, Jakarta, Indonesia

* marcel71@ui.ac.id, prasetyo.ui.ac@gmail.com

**Data Availability Statement:** The data underlying the results presented in the study are available

## Abstract

### Background

Hemophilic arthropathy, a condition manifested as joint destruction due to spontaneous joint bleeding, is one complication of hemophiliac patients. Early detection and intervention may improve the outcome, in which ultrasonography can be an ideal modality with the introduction of HEAD-US (Hemophilia Early Arthropathy Detection with Ultrasound) protocol. Studies have shown US benefit in hemophiliac patients, including its potential as an alternative for the Hemophiliac Joint Health Score (HJHS) system. However, many of the studies were conducted in countries with better management of hemophilia using prophylaxis treatment. It is unclear whether HEAD-US has a correlation with HJHS in countries using episodic treatment only, like in Indonesia.

### Purpose

This study aimed to explore the correlation between HEAD-US and HJHS in hemophiliac patients with joint problems in Indonesia.

### Materials and methods

A cross-sectional correlation study between HEAD-US and HJHS was performed with primary data collected from 120 hemophilic patients. US examination was performed on elbow, knee and ankle joints using the HEAD-US scoring method by a musculoskeletal radiologist. HJHS examination was conducted by a trained physiotherapist and a medical rehabilitation specialist. All examiner is member of multidisciplinary Hemophiliac Management Team in Cipto Mangunkusumo General Hospital in Jakarta, Indonesia.

from the corresponding author and Ethics Committee for researchers who meet the criteria for access to confidential data (Ethics Committee of Faculty Medicine Universitas Indonesia and The Center for Clinical Epidemiology and Evidence-Based Medicine (CEEBM) FKUI-RSCM. (Contact for Ethics Committee of Faculty Medicine Universitas Indonesia: ec_fkui@yahoo.com or +62213157008).

**Funding:** The authors received no specific funding for this work.

**Competing interests:** The authors have declared that no competing interests exist.

**Abbreviations:** HEAD-US, Hemophilia Early Arthropathy Detection with Ultrasound; HJHS, Hemophilia Joint Health Score; USG, Ultrasonography.

## Results

The mean age of the participant was 9.3 (5–14) years old. The median score of HEAD-US was 8 (1–28) with most of the joint abnormalities found on the ankles. The median score of HJHS was 3 (0–35), with most joint abnormalities found on the knees. There was a moderate correlation between HEAD-US and HJHS score ($p < 0.05$, $r = 0.65$).

## Conclusion

HEAD-US shows a moderate correlation to HJHS in hemophiliac patients who received episodic treatment. HEAD-US can provide additional value in the anatomical evaluation of the joint and could be complementary to HJHS in assessing the joint status in hemophilic patient

## Introduction

Hemophilia is a genetically hereditary coagulopathy disease caused by factor VIII (Hemophilia A) or factor IX (hemophilia B) coagulation deficiency that occurs in 1 out of 10000 people [1–3]. Joint bleeding is one of the most frequent complications found mostly in the elbow, knee and ankle [4, 5]. Several modalities are used to detect these complications including radiographic and MR examination. Besides, the use of ultrasonography (US) imaging has been increasing due to its practicality, relatively low cost, and can be repeated easily in the clinical setting. US can evaluate early blood-induced joint changes such as synovial hypertrophy and cartilage defect [6–8]. Unfortunately, the US result highly depends on the operator's skill [9–11]. A study by Martinoli et al. [12] has introduced a standardized, fast, repeatable, and simple US scoring protocol for hemophiliac joint called *Hemophilia Early Arthropathy Detection with Ultrasound* (HEAD-US). This standardized method can assess multiple joints simultaneously with an excellent inter-observer agreement.

A study by Foppen et al. [13] analyzed the correlation between HEAD-US scoring and Hemophilia Joint Health Score (HJHS), a clinical method to evaluate functional status in the hemophilic joint. This study showed a strong correlation between the HEAD-US method and HJHS scoring. However, this study was conducted on hemophilia patients who had received prophylaxis treatment. In Indonesia, most hemophiliac patients receive episodic treatment, due to financial challenges and unavailability of prophylaxis treatment [14, 15]. Therefore, many pediatric hemophiliac patients have moderate to severe joint problems. Since HEAD-US is designed to assess early hemophilic joint changes, it is not known whether it will also perform well in moderate or severe stages of joint disease, especially if we considering its correlation to HJHS. This study aimed to evaluate the correlation between ultrasonography findings using the HEAD-US method and HJHS assessment scale on evaluating the musculoskeletal status of hemophilic arthropathy patients.

## Materials and methods

### Participants and study protocol

A cross-sectional correlation study was conducted in the Department of Radiology and Department of Physical and Rehabilitation Medicine Faculty of Medicine, Universitas Indonesia in Cipto Mangunkusumo National Hospital Jakarta from June until September 2016. Subjects were patients with severe hemophilia A without any sign of active joint bleeding, history

of inhibitor, history of musculoskeletal surgery, tumor and trauma of the joints. A total of 20 subjects were included in the study and written informed consent was obtained from the parents and legal guardian. The study was approved by the Research Ethical Committee of the Faculty of Medicine Universitas Indonesia (No.633/UN2.F1/ETIK/2016).

## Musculoskeletal assessment

US examination on the elbow, knee and ankle joints using HEAD-US protocol was performed by a radiologist consultant in musculoskeletal with at least 10-years of experience in musculo-skeletal ultrasound. The examination was performed using Samsung Medison US equipment with linear array 11 MHz transducer.

HJHS examination version 2.1 was conducted by a trained physiotherapist and a medical rehabilitation specialist. All examiner is member of multidisciplinary Hemophiliac Management Team in Cipto Mangunkusumo General Hospital in Jakarta, Indonesia. A total of 17 patients underwent the US examination before the HJHS examination and 3 patients underwent the HJHS examination before the US examination. Both examinations were blinded and performed on the same day.

## Statistical analysis

The quantitative data were analyzed statistically using Statistical Package for Social Sciences (SPSS) version 20.0 software. The normality test was conducted with the Shapiro-Wilk test showed abnormal data distribution; therefore, the bivariate analysis using Spearman's correlations test was done to analyze the correlation between HEAD-US and HJHS examination. The value of the correlation coefficients (r) was further classified into 0.0–0.1 indicating 'no correlation'; 0.1–0.3 indicating 'poor correlation'; 0.3–0.6 indicating 'fair correlation'; 0.6–0.8 indicating 'moderate correlation'; 0.8–0.9 indicating 'very strong correlation'; and 0.9–1.0 indicating 'perfect correlation' [16].

# Results

## Characteristic of studied individuals

Twenty subjects were analyzed with ages ranged from 5 to 14 years old (mean age 9,35 years old). All subjects received episodic treatment. Positive gait sign was noticed in 4 subjects, as shown in **Table 1**.

**Table 1. Characteristic of subjects.**

| Characteristic of subjects | Frequency (N) | Percentage (%) |
|---|---|---|
| Age (years) | | |
| < 10 | 12 | 60 |
| >10 | 8 | 40 |
| Degree of hemophilia | | |
| Severe | 20 | 100 |
| Mild | 0 | 0 |
| Treatment History | | |
| On Demand | 20 | 100 |
| Prophylaxis | 0 | 0 |
| Gait sign | | |
| Positive | 4 | 20 |
| Negative | 16 | 80 |

**Table 2. The results of HJHS and HEAD-US examination.**

| Score | Mean ± SD | Median | Min | Max |
|---|---|---|---|---|
| HEAD-US | 9.9 ± 6.7 | 8.0 | 1 | 28 |
| HJHS | 4.7 ± 7.6 | 3.0 | 0 | 35 |

From a total of 120 joints examined, the HEAD-US scored between 1–28 with a median score of 8. Generally, the ankle joint was found to be affected more severe compared to the elbow and knee joint. HJHS examination score ranged between 0–35 with a median score of 3 (**Table 2** and **Fig 1**).

Spearman's test showed a statistically significant correlation between HEAD-US and HJHS with a moderate positive correlation (p-value <0.05 and r = 0.652) as shown in **Fig 2**.

Based on linear regression analysis, a linear model was obtained to predict the HJHS score based on the HEAD-US score (**Eq 1**).

$$\text{HJHS score} = -3,74 + 0,86 \text{ x HEAD-US} \qquad \text{Eq1}$$

## Discussion

US examination using the HEAD-US method showed that the most affected joint in this study was the ankle joint. This finding was consistent with the study conducted by Oldenburg et al. [17], Altisent et al. [18] and Aisa et al. [19], which stated the ankle joint as the main target in hemophilic arthropathy before knee joint destruction happened. Early arthropathic changes can be detected with synovial hypertrophy was the most common abnormality [19]. These findings were different from the older study by Knobe and Berntorp [20], which stated that the knee joint was the most affected target joint in severe hemophilia patients who had not received any prophylaxis treatment. US examination has high sensitivity in detecting early joint changes underlying the reason why ankle abnormalities can be detected more frequently than others.

Most of the joints (57 out of 120 joint) showed grade 1 synovial hypertrophy on US examination. Some of this joint change was noted in the joint that had no history of previous bleeding. Additionally, 46 of 120 joints (38%) showed abnormalities in HEAD-US while the HJHS score remained zero. Joint bleeding likely happened in a quite small amount result in the subclinical sign that can be undetected clinically and did not result in functional disability. The ability of the US in detecting early anatomical changes before any observed clinical signs were also found in a study by Oldenburg et al. [17] Similar findings were also found by Altisent, et al. [18] with 19.8% and by Timmer, et al. [21] with 18.4% of patient showing abnormalities only in the HEAD-US. However, this study showed a higher percentage compared to the previous studies due to no prophylaxis treatments were given before.

Only 10 of 120 joints (8.3%) have positive HJHS while the HEAD-US score is 0. According to Slutier, et al. [22], HJHS positive findings such as flexion loss and crepitus without clinical complaint can be found in minor injury or anatomical variants other than the hemophilic joint. The author suggests that the HJHS score of up to 3 points should be considered as 'normal'. In our study, subjects with 0 score on the HEAD-US and positive HJHS findings showed scores not exceeding 3 points.

Further analysis on each joint of subjects with an HJHS score of 0 showed higher positive HEAD-US finding in 21 of 40 ankle joint (67.5%), in comparison with knee joint (25%) and elbow joint (22.5%). The ankle joint may be the first joint affected due to weight-bearing activity, thus results in more anatomical changes than the knee and elbow joint.

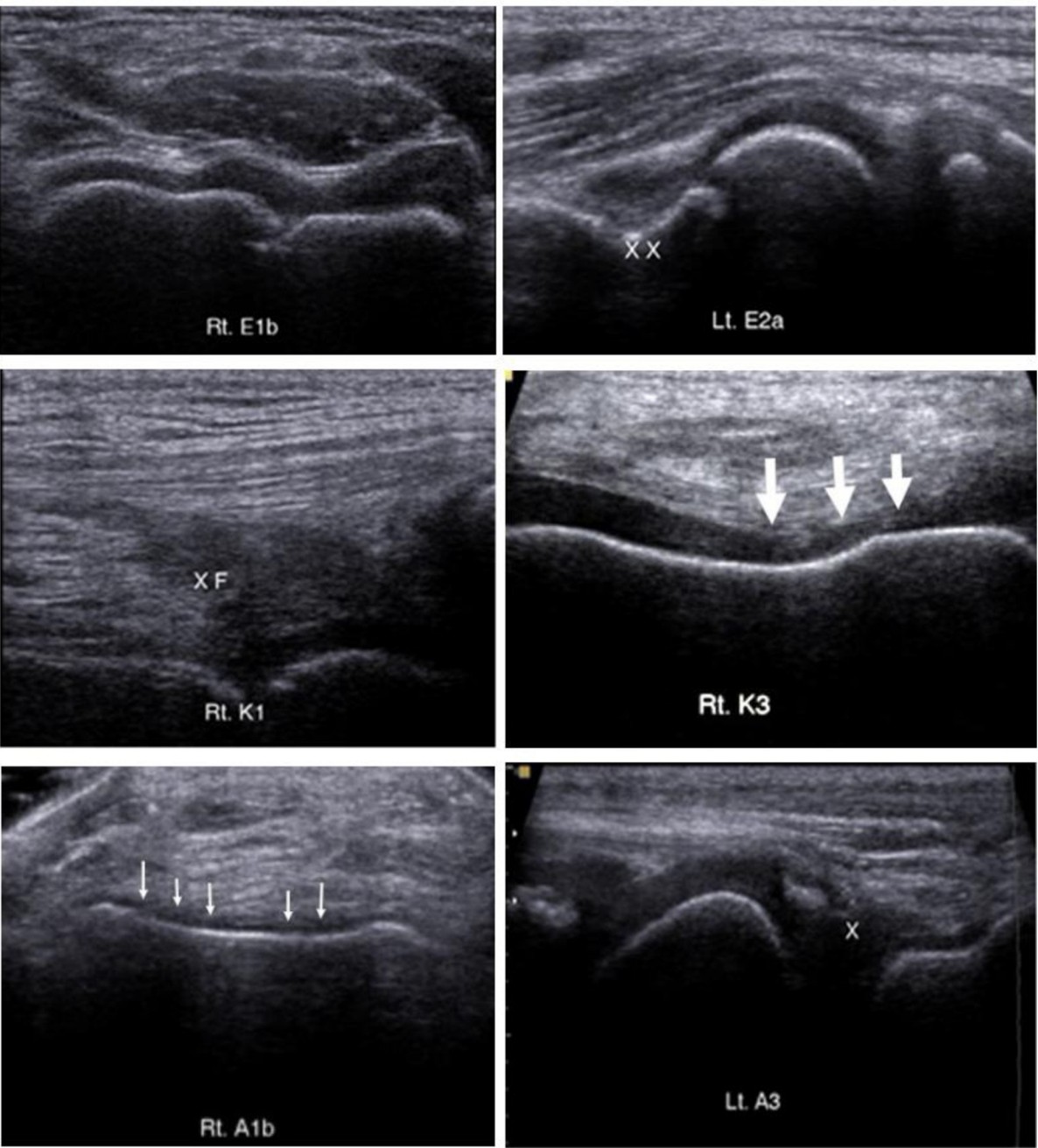

**Fig 1. The example of HEAD-US appearance on different subjects: Elbow (E1b, E2a), knee (K1, K3), and ankle (A1b, A3).** There is synovial hypertrophy of the anterior cubital fossa of the elbow (XX), the suprapatellar recess of the knee (XF) and the posterior of the subtalar joint (X). There is also cartilage irregularity and thinning at the femoral trochlea of the knee (large arrow) and talar dome of the ankle (small arrow).

Spearman's correlation analysis showed a moderate correlation between HEAD-US and HJHS score for all joints (p-value < 0.05 and r value = 0.652). This result was almost similar to the study by Foppen et al. [13] which found a significant correlation coefficient of 0.7. Another study by De La Corte-Rodriguez, et al. [23] found a correlation coefficient of 0.717 and an even higher correlation was found by Timmer, et al. [21] with a correlation coefficient of 0.88.

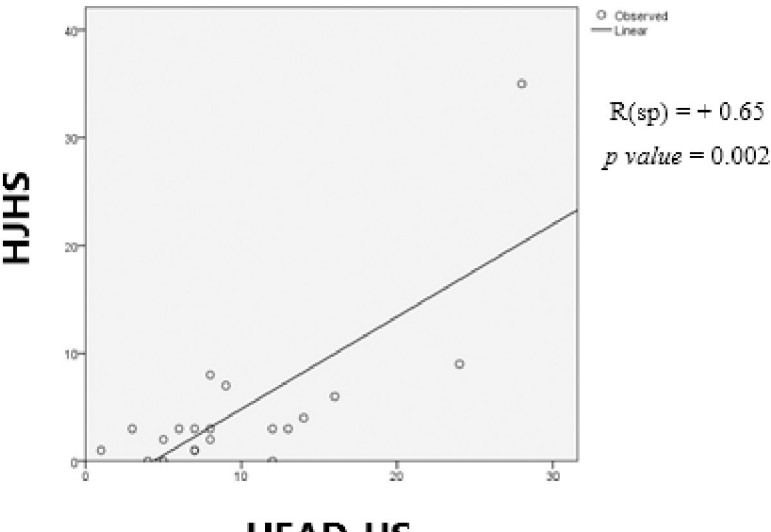

**Fig 2. Correlation between HEAD-US and HJHS.**

This difference is probably due to the different number of abnormalities that were found in the involved individuals. Foppen et al. [13] found abnormalities in 5 out of 63 joints in patients receiving prophylaxis treatment, while this study on patients with episodic therapy found abnormalities in 73 out of 120 joints based on HEAD-US and 37 of 120 joints based on HJHS. Linear regression analysis developed a linear model to predict the HJHS score based on the HEAD-US score (**Eq 1**).

Further analysis on each joint (elbow, knee, ankle) found a better correlation strength between HEAD-US and HJHS in knee joint abnormality (r = 0.51). On the other hand, correlation strength was lower in the elbow joint (r = 0.4) and ankle joint (r = 0.36).

There were limitations regarding to this study. S*ubjects were limited to the pediatric patients due to the limitation of the Multidisciplinary Hemophiliac Management Team of the hospital's database which covering age group from 5 to 18 years old which make these age groups become the most available and met the inclusion criteria.* HEAD-US examination was performed by a single radiologist experienced in MSK ultrasound and US was an operator dependent modality. Although previous studies had shown an excellent inter-observer agreement, it probably would be more objective if performed by more than one operator.

## Conclusion

HEAD-US shows a moderate correlation to HJHS in hemophiliac patients who received episodic treatment. HEAD-US may provide additional value in detecting anatomical joint changes before any abnormalities are observed clinically, while HJHS is better in evaluating functional status. HEAD-US and HJHS will likely become more beneficial if both are performed as a complement to each other in assessing the patient with hemophilic arthropathy.

## Acknowledgments

The authors wish to thank the Department of Radiology, Department of Medical Rehabilitation and Departement of Community Medicine Faculty of Medicine Universitas Indonesia for the contribution on the facilities for the assessment of HEAD-US and HJHS.

## Author Contributions

**Conceptualization:** Marcel Prasetyo, Ratna Moniqa, Angela Tulaar, Joedo Prihartono, Stefanus Imanuel Setiawan.

**Data curation:** Marcel Prasetyo, Ratna Moniqa, Angela Tulaar, Joedo Prihartono, Stefanus Imanuel Setiawan.

**Formal analysis:** Marcel Prasetyo, Ratna Moniqa, Angela Tulaar, Joedo Prihartono, Stefanus Imanuel Setiawan.

**Funding acquisition:** Marcel Prasetyo.

**Investigation:** Marcel Prasetyo, Ratna Moniqa, Angela Tulaar.

**Methodology:** Marcel Prasetyo, Ratna Moniqa, Angela Tulaar, Joedo Prihartono, Stefanus Imanuel Setiawan.

**Project administration:** Marcel Prasetyo, Ratna Moniqa.

**Resources:** Marcel Prasetyo, Ratna Moniqa.

**Software:** Marcel Prasetyo, Joedo Prihartono, Stefanus Imanuel Setiawan.

**Supervision:** Marcel Prasetyo, Angela Tulaar, Joedo Prihartono.

**Validation:** Marcel Prasetyo, Angela Tulaar, Joedo Prihartono, Stefanus Imanuel Setiawan.

**Writing – original draft:** Ratna Moniqa, Stefanus Imanuel Setiawan.

**Writing – review & editing:** Marcel Prasetyo, Stefanus Imanuel Setiawan.

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
