## [Decision Letter · Decision Letter 0]

13 Nov 2020

PONE-D-20-29192

Correlation between Hemophilia Early Arthropathy Detection with Ultrasound (HEAD-US) Score and Hemophilia Joint Health Score (HJHS) in Patients with Hemophilic Arthropathy

PLOS ONE

Dear Dr. Prasetyo,

Thank you for submitting your manuscript to PLOS ONE. After careful consideration, we feel that it has merit but does not fully meet PLOS ONE’s publication criteria as it currently stands. Therefore, we invite you to submit a revised version of the manuscript that addresses the points raised during the review process.

Both reviewers felt that this paper showed some merit although it would benefit from minor corrections. A concern was that the population sampled (adolescents) may not be the highest risk for hemarthropathy and the care described (episodic rather than prophylactic factor replacement) no longer represent standard of care. In addition, the minor points raised regarding the use of portable ultrasonography should be considered.

We look forward to receiving your revised manuscript.

Kind regards,

Elizabeth S. Mayne, M.D.

Academic Editor

PLOS ONE

Journal Requirements:

Additional Editor Comments:

Both reviewers felt that this paper showed some merit although it would benefit from minor corrections. A concern was that the population sampled (adolescents) and the care described (episodic rather than prophylactic factor replacement) no longer represent standard of care. In addition, the minor points raised regarding the use of portable ultrasonography should be considered.

Reviewers' comments:

Reviewer's Responses to Questions

**Comments to the Author**

1. Is the manuscript technically sound, and do the data support the conclusions?

Reviewer #1: Yes

Reviewer #2: Yes

2. Has the statistical analysis been performed appropriately and rigorously? 

Reviewer #1: No

Reviewer #2: Yes

3. Have the authors made all data underlying the findings in their manuscript fully available?

Reviewer #1: Yes

Reviewer #2: Yes

4. Is the manuscript presented in an intelligible fashion and written in standard English?

Reviewer #1: Yes

Reviewer #2: Yes

5. Review Comments to the Author

Reviewer #1: Correlation between Hemophilia Early Arthroplasty Detection with Ultrasound (HEADUS) Score and Hemophilia Joint Health Score (HJHS) in Patients with Hemophilic Arthropathy

In this single-centre, cross-sectional study, the authors evaluated the correlation between the HJHS and HEAD-US in twenty pediatric and adolescent patients with severe hemophilia using already established protocols. A trained physiotherapist evaluated only elbows, knees and ankles. The ultrasound and HJHS were done the same day of the visit and showed a correlation of 0.65.

Although this work has already been performed elsewhere, validation of this correlation in the setting of the authors geographic location may add value to the existing body of knowledge. This report could be improved by addressing the following queries.

1. The introduction should include the rationale for doing this correlation assessment in patients receiving episodic treatment when it is not the standard of care in hemophilia in 2020 anymore. How is this knowledge going to be translated into clinical practice?

2. The study population is adolescent and pediatric patients who often have minimum arthropathy. Is there a rationale for doing the study in this group only and not in adults?

3. The researchers should include the standard shortcomings of ultrasound examination in the introduction or discussion as the study shortcomings.

4. Given that the ultrasound was done in a hospital setting and the expanding role of portable ultrasound, the authors should point out the role of portable ultrasound in their setting.

5. In the methods, it is unclear how the physiotherapists/ rehabilitation specialists were trained and what their intra-individual and inter-individual coefficient of variation was.

6. The discussion should include the strengths and weaknesses of the study.

7. In the abstract, reference is made to joint bleed and hemarthrosis, which mean the same thing.

8. The reference page numbers should be reviewed for consistency, some have single digit and others have two digit last page numbers.

9. Can the authors comment on their statement that a correlation of 0.65 is regarded as a strong correlation in hemophilia?

Reviewer #2: It is my opinion that the authors have demonstared a relatively easy reproducible way of assesng joint involvement in Hemophilia and the paper should be published

6. PLOS authors have the option to publish the peer review history of their article (what does this mean?). If published, this will include your full peer review and any attached files.

Reviewer #1: **Yes: **Johnny Mahlangu

Reviewer #2: **Yes: **BF Jacobson

---

## [Author Response · Author response to Decision Letter 0]

6 Jan 2021

RE: ACADEMIC REBUTTAL LETTER

Dear PLoS One Editors and Reviewers,

As you are aware, I have submitted a research manuscript titled “Correlation between Hemophilia Early Arthropathy Detection with Ultrasound (HEAD-US) Score and Hemophilia Joint Health Score (HJHS) in Patients with Hemophilic Arthropathy” in the online editorial manager of the PLoS One a while ago. The manuscript needs revision which mainly focuses on:

- Adding specific information on the rationale for doing this correlation assessment in patients receiving episodic treatment when it is not the standard of care in hemophilia in 2020, the rationale for doing the study in adolescent and paediatric patients group only and not in adults, how the physiotherapists/ rehabilitation specialists were trained and what their intra-individual and inter-individual coefficient of variation was and further explanation of the strengths and weaknesses of the study.

- Further correction on the reference page numbers, abstract, and degree of correlation in results and discussion.

Besides, we would also answer the reviewer questions related to the study stated in the comments to the author:

1. The introduction should include the rationale for doing this correlation assessment in patients receiving episodic treatment when it is not the standard of care in hemophilia in 2020 anymore. How is this knowledge going to be translated into clinical practice? 

Answer: we already revised the introduction regarding the reason why the subjects receiving episodic treatment. In Indonesia, most hemophiliac patients receive episodic treatment, due to financial challenges and unavailability of prophylaxis treatment. 

2. The study population is adolescent and pediatric patients who often have minimum arthropathy. Is there a rationale for doing the study in this group only and not in adults?

Answer: There were not many studies that include pediatric subjects exclusively. Besides, subjects were limited to the pediatric patients due to the limitation of the Multidisciplinary Hemophiliac Management Team of the hospital’s database which covering age group from 5 to 18 years old which make these age groups become the most available and met the inclusion criteria.

3. The researchers should include the standard shortcomings of ultrasound examination in the introduction or discussion as the study shortcomings.

Answer: The shortcoming of US was operator dependent and already mentioned in the limitation of the study in discussion. 

4. Given that the ultrasound was done in a hospital setting and the expanding role of portable ultrasound, the authors should point out the role of portable ultrasound in their setting.

Answer: The portable US was not widely available in Indonesia. However, if it is available, the portable US could be one alternative modality in order to assess the HEAD-US.

5. In the methods, it is unclear how the physiotherapists/ rehabilitation specialists were trained and what their intra-individual and inter-individual coefficient of variation was.

Answer: There were no data regarding intra and inter-individual coefficient of variation between the specialists.

6. The discussion should include the strengths and weaknesses of the study.

Answer: Already added in the discussion section.

7. In the abstract, reference is made to joint bleed and hemarthrosis, which mean the same thing.

Answer: Already revised.

8. The reference page numbers should be reviewed for consistency, some have single digit and others have two digit last page numbers.

Answer: Already revised.

9. Can the authors comment on their statement that a correlation of 0.65 is regarded as a strong correlation in hemophilia?

Answer: Already revised. (Moderate)

We also remove the figures/ from within your manuscript file and mention table 2 in the text.

Based on these suggestions, we have revised the manuscript according to the reviews by adding additional information on the introduction, methods, results and discussion section. Please find the attached revised manuscript and figure file in the online platform. Please address all correspondence concerning this manuscript to me at prasetyo.ui.ac@gmail.com.

The data underlying the results presented in the study are available from the corresponding author and need the Ethics Committee approval (there are ethical or legal restrictions on sharing a de-identified data set, due to potentially identifying or sensitive patient information) for researchers who meet the criteria for access to confidential data (Ethics Committee of Faculty Medicine Universitas Indonesia and The Center for Clinical Epidemiology and Evidence-Based Medicine (CEEBM) FKUI-RSCM (contact for Ethics Committee of Faculty Medicine Universitas Indonesia: ec_fkui@yahoo.com or +62213157008)

Thank you for your consideration of this manuscript. 

Sincerely,

Marcel Prasetyo, M.D., Ph.D.

---

## [Decision Letter · Decision Letter 1]

9 Mar 2021

Correlation between Hemophilia Early Arthropathy Detection with Ultrasound (HEAD-US) Score and Hemophilia Joint Health Score (HJHS) in Patients with Hemophilic Arthropathy

PONE-D-20-29192R1

Dear Dr. Prasetyo,

We’re pleased to inform you that your manuscript has been judged scientifically suitable for publication and will be formally accepted for publication once it meets all outstanding technical requirements.

Kind regards,

Elizabeth S. Mayne, M.D.

Academic Editor

PLOS ONE

Additional Editor Comments (optional):

Reviewers' comments:

Reviewer's Responses to Questions

**Comments to the Author**

1. If the authors have adequately addressed your comments raised in a previous round of review and you feel that this manuscript is now acceptable for publication, you may indicate that here to bypass the “Comments to the Author” section, enter your conflict of interest statement in the “Confidential to Editor” section, and submit your "Accept" recommendation.

Reviewer #1: All comments have been addressed

2. Is the manuscript technically sound, and do the data support the conclusions?

Reviewer #1: Yes

3. Has the statistical analysis been performed appropriately and rigorously? 

Reviewer #1: Yes

4. Have the authors made all data underlying the findings in their manuscript fully available?

Reviewer #1: Yes

5. Is the manuscript presented in an intelligible fashion and written in standard English?

Reviewer #1: Yes

6. Review Comments to the Author

Reviewer #1: (No Response)

7. PLOS authors have the option to publish the peer review history of their article (what does this mean?). If published, this will include your full peer review and any attached files.

Reviewer #1: **Yes: **Johnny Mahlangu

---

## [Editor Report · Acceptance letter]

11 Mar 2021

PONE-D-20-29192R1 

Correlation between Hemophilia Early Arthropathy Detection with Ultrasound (HEAD-US) Score and Hemophilia Joint Health Score (HJHS) in Patients with Hemophilic Arthropathy 

Dear Dr. Prasetyo:

I'm pleased to inform you that your manuscript has been deemed suitable for publication in PLOS ONE. Congratulations! Your manuscript is now with our production department. 

Kind regards, 

on behalf of

Dr. Elizabeth S. Mayne 

Academic Editor

PLOS ONE